# Development of a Novel Three Degrees-of-Freedom Rotary Vibration-Assisted Micropolishing System Based on Piezoelectric Actuation

**DOI:** 10.3390/mi10080502

**Published:** 2019-07-29

**Authors:** Yan Gu, Xiuyuan Chen, Faxiang Lu, Jieqiong Lin, Allen Yi, Jie Feng, Yang Sun

**Affiliations:** 1School of Mechatronic Engineering, Changchun University of Technology, Changchun 130012, China; 2Department of Industrial, Welding and Systems Engineering, Ohio State University, Columbus, OH 43210, USA; 3Changchun Equipment and Technology Research Institute, Norinco Group, Changchun 130012, China

**Keywords:** nonresonant micropolishing, vibration-assisted processing device (VPD), silicon carbide (SiC) ceramic, finite element analysis

## Abstract

The limited degrees of freedom (DOF) and movement form of the compliant vibration-assisted processing device are inherent constraints of the polishing technique. In this paper, a concept of a 3-DOF rotary vibration-assisted micropolishing system (3D RVMS) is proposed and demonstrated. The 3-DOF means the proposed vibration-assisted polishing device (VPD) is driven by three piezo-electric (PZT) actuators. Compared with the current vibration-assisted polishing technology which generates a trajectory with orthogonal actuators or parallel actuators, a novel 3-DOF piezoelectrically actuated VPD was designed to enable the workpiece to move along the rotational direction. Meanwhile, the proposed VPD can deliver large processing stoke in mrad scale and can be operated at a flexible non-resonant mode. A matrix-based compliance modeling method was adopted for calculating the compliance and amplification ratio of the VPD. Additionally, the dynamic and static properties of the developed VPD were verified using finite element analysis. Then, the VPD was manufactured and experimentally tested to investigate its practical performance. Finally, various polished surfaces which used silicon carbide (SiC) ceramic as workpiece material were uniformly generated by the high-performance 3D RVMS. Compared with a nonvibration polishing system, surface roughness was clearly improved by introducing rotary vibration-assisted processing. Both the analysis and experiments verified the effectiveness of the present 3D RVMS for micro-machining surfaces.

## 1. Introduction

For optical systems to continually exhibit high hardness, strength, and light weight, SiC ceramic is an ideal material for building space-based optical information collection systems. It also has inherent properties such as low thermal distortion, toxicity, and potential cost relative to other conventional materials [1,2]. However, the processing of SiC into a practical surface topography presents a challenge resulting from its high hardness and brittleness [3,4]. To realize the ideal tolerance and surface quality, diamond abrasive processing of SiC through computer numerical control (CNC) polishing is one of the major technologies.

Traditionally, the CNC polishing method was more widely utilized for manufacturing optics. Motivated by a nondeterministic method, Jones et al. proposed a machine for manufacturing large aspheric optics with CNC polishing through interferometric inspection [5]. To accelerate the fabrication of mirrors which are utilized in large telescopes, Yang et al. used CNC combined with a proposed test device to polish and detect the remaining surface error [6]. Considering deterministic finishing, Hu et al. studied the principles based on residual errors to achieve corrective polishing [7].

Although conventional polishing processes such as mechanical polishing still use as universal precision-machining technologies, the required polishing accuracy and processing efficiency are difficult to achieve. The wear and rapid blunting of polishing tools also reduce the operating life and increase the costs. To solve these defects, many scholars have paid close attention to the field of ultrasonic-assisted polishing. Suzuki et al. introduced ultrasonic vibration into polishing aspheric molds to obtain large numerical apertures (NAs) [8,9]. To achieve material removal effectively, a new elliptical ultrasonic-assisted grinding method was presented by Liang et al., whose experimentally validated surface roughness was reduced by 10% compared with the conventional grinding [10]. Zhao et al. revealed that ultrasonic vibration could reduce the friction force in processing SiC cylindrical surfaces, which resulted in enhanced surface integrity through experimental and calculation results [11]. Compared to the travel scale of which most resonant vibration-assisted devices are micrometer, the proposed vibration-assisted polishing device (VPD) can deliver maximum rotational angle in mrad scale because of 3-degrees of freedom (DOF) piezoelectrically actuating. It is true that most resonant mechanisms are suffering from the drawbacks of small vibration amplitudes [8,9,10,11].

According to the aforementioned efforts, it is obvious to conclude that vibration-assisted machining has shown its capacity for improving machinability [12]. Further, resonant and nonresonant are the major operating modes of vibration-assisted systems. The resonant system has the advantage of using energy efficiently; however, it can only operate at certain frequencies. By contrast, the nonresonant system is generally designed with piezoelectric-actuated flexure structures and can operate at continuous frequencies. Considering the flexibility of the polishing process, it is suitable to develop a processing system through introducing nonresonant vibration. Continuous efforts to increase the machining ability through applied nonresonant vibration assistance have been conducted, and a significant number of breakthroughs have been reported by many researchers [13,14]. Based on the concept of 2-DOF FTS-assisted diamond turning, Zhu et al. introduced Z-shaped flexure hinges as a novel guidance structure to realize the fabrication of scattering homogenization surfaces [15]. Considering the influence of vibration-assisted micromilling quality, the desired vibration was generated from the aluminum alloy Al 6061-T6 on a 2-DOF vibration worktable proposed by Chern et al. [16].

Regarding the limited nature of nonresonant vibration-assisted processing systems, the orthogonal or parallel vibration which are actuated by 2-DOF piezo-electric (PZT) actuators was recently applied for improving the capability of generating targeted micro/nanostructured textures. For example, Zhu et al. developed a novel parallel-kinematic configuration which was actuated by 2-DOF PZT actuators to achieve accurate surface textures [17]. Compared to the current nonresonant VPD generating trajectory through 2-DOF PZT actuators, our target is to design a novel 3-DOF piezoelectrically actuated vibration-assisted method to enable the workpiece to move along the rotational direction.

Based on the above discussion, a novel 3D rotary vibration-assisted micropolishing system (RVMS) based on piezoelectric actuation is a suitable choice. The compliance matrix method was adopted to establish theoretical models of the VPD in terms of compliance and the amplification ratio. In addition, the dynamic and static performances of the proposed VPD were determined based on finite element analysis (FEA). Meanwhile, both open-loop and closed-loop experimental tests were conducted to determine the properties of the VPD. Finally, high-quality SiC ceramic surfaces were generated by 3D RVMS in order to validate the practical machining performance.

This paper is arranged as follows: the principle of the designed 3D RVMS is presented in Section 2. Based on the compliance matrix method, static modeling is conducted in Section 3, where the compliance and amplification ratio of the VPD are analyzed. In Section 4, FEA and experimental tests are implemented on the fabricated prototype to test the properties of the proposed VPD. Finally, an on-machine performance experiment of the 3D RVMS is conducted in Section 5. The conclusion is drawn in Section 6.

## 2. Design of Proposed 3-DOF Rotary Vibration-Assisted Micropolishing System (3D RVMS)

### 2.1. Motivation for Vibration-Assisted Mechanism

Figure 1a shows the major moving parts of the VPD. In the view of kinematics, the flexure hinges can generate rotational motion along a certain axis and serve as the desired revolute joints. Thus, a pseudo-rigid body is established in Figure 1b. *A_ij_* (*i*, *j* = 1, 2, and 3) represents the revolute joints of the links connecting the stationary frame, *B_ij_* (*i*, *j* = 1, 2, and 3) denotes the revolute joints between the two active links, and *C_i_* (*i* = 1, 2, and 3) represents the revolute joints that connect the active links to the center stage. When the piezoelectric forces *F_1_*, *F_2_*, and *F_3_* are exerted to the input ends, respectively, the linear motions of *B_11_B_31_*, *B_12_B_32_*, and *B_13_B_33_* are transferred into the ideal rotary motion of the output center stage. It is noted that the revolute-joint offset between the relative links is ignored.

### 2.2. Principle of the Proposed 3D RVMS

To realize a gentle removal process, fine abrasive slurry is carried by the fluid, as shown in Figure 2. The polishing system consists of a VPD, low-frequency-driven actuators, and a polishing tool. A CNC positions the relatively hard polyurethane polishing tool. When the workpiece is actuated by the VPD at an in-plane rotational angle, the workpiece circulates the fine slurry, which creates shear forces between the workpiece and the polishing tool. Accordingly, the workpiece is polished by the shear forces based on the vibration amplitude and the frequency.

A generalized material removal equation is derived from the well-known Preston’s equation
(1)$$h=C_{p}v_{s}F_{N}\Delta t$$
where *C_p_* is a constant, which is influenced prominently by the material; *v_s_* is the magnitude of the relative speed; $F_{N}$ is the normal polishing force; and Δt is the dwell time. The workpiece rotational speed $v_{w}$ is given by Equation (2) below
(2)$$v_{w}=2\pi Af\cdot Ramp$$
where *Ramp* is the amplification ratio of the VPD, *A* and *f* represent the amplitude and frequency of the PZT actuators, respectively. Accordingly, $v_{s}$ can be derived as
(3)$$v_{s}=\sqrt{{\left(2\pi w_{t}r\right)}^{2}+v_{w}{}^{2}}$$
where $w_{t}$ is the polishing tool rotational angular velocity, $v_{w}$ is the VPD rotational speed, and *r* is the polishing tool radius. *v_s_* can be simplified as $v_{s}\approx 2\pi w_{t}r$ because $2\pi w_{t}r>>v_{w}$. From this, it is concluded that the rotation of the VPD is not the major factor in improving the removal depth or the volumetric removal rate of the workpiece.

However, according to the crest from the surface, it has been removed through introducing vibration-assisted by the alternating cycle phenomenon [18]. Thus, it is assumed that a high-quality surface can be achieved by the vibratory workpiece. Further, vibration makes the workpiece remove the surface peaks and valleys by moving the polishing tool along the processing area from the prior cycle. This assumption will be verified by the processing experiments in Section 5.

## 3. Compliance and Amplification Ratio Analysis Based on the Matrix Method of the VPD

It must be said that stiffness and output rotational angle (related to the amplification ratio) are important factors that should be considered for guiding the micro-polishing experiments. Firstly, the stiffness can be classified into input stiffness and output stiffness. In the processing experiments, the VPD’s input stiffness must be less than the actuators’ static stiffness and high output stiffness is used to resist the polishing forces. Moreover, it is important to investigate the size of workspace (maximum rotational angle) of the proposed VPD because it is related to the ability of improving the surface quality. Some recently developed processing methods are described by introducing the modelling and testing of static and dynamic properties of vibration-assisted devices [17,19].

### 3.1. Output Compliance Modeling

The static performance of the VPD is associated with the hinges. Accordingly, right-angle hinges and right-circular hinges are the most suitable choices (as shown in Figure 3), owing to their advantage of high accuracy. Since the VPD is planar with a negligible out-of-plane stiffness, both hinges can be simplified into 3-DOF [20,21]. In addition, since the related displacement of point *O_i_* is $D={\left[\begin{array}{ccc} u_{x} & u_{y} & \theta_{z} \end{array}\right]}^{T}$ when a load vector $\;F={\left[\begin{array}{ccc} F_{x} & F_{y} & M_{z} \end{array}\right]}^{T}$ of point *O_i_* is applied in/around certain axes, the deformation equation is derived based on Hook’s law
(4)$$\left[\begin{array}{c} u_{x}\\ u_{y}\\ \theta_{z} \end{array}\right]=\left[\begin{array}{ccc} C_{u_{x}-F_{x}} & 0 & 0\\ 0 & C_{u_{y}-F_{y}} & C_{u_{y}-M_{z}}\\ 0 & C_{\theta_{z}-F_{y}} & C_{\theta_{z}-M_{z}} \end{array}\right]\left[\begin{array}{c} F_{x}\\ F_{y}\\ M_{z} \end{array}\right]$$
where the compliance factors can be obtained through Young’s modulus of the material and the hinge’s dimensional parameters, as shown in Figure 3 [22,23].

Transforming *O_i_-xy* from local coordinates to the target coordinates *O_j_-xy*, the transformation matrix $T_{i}^{j}$ can be written as
(5)$$T_{i}^{j}={\bar{R}}_{i}^{j}{\bar{P}}_{i}^{j}=\left[\begin{array}{ccc} r_{11} & r_{12} & r_{13}\\ r_{21} & r_{22} & r_{23}\\ r_{31} & r_{32} & r_{33} \end{array}\right]\left[\begin{array}{ccc} 1 & 0 & P_{y}\\ 0 & 1 & -P_{x}\\ 0 & 0 & 1 \end{array}\right]$$
where ${\bar{R}}_{i}^{j}$ is the rotational matrix, and ${\bar{P}}_{i}^{j}$ represents the translational matrix.

The kinematic chains for a third of the VPD, as shown in Figure 4a, can be demonstrated as three springs connected in parallel at point *O* (Figure 4b) [24]. According to the serial connection of hinges 1 and 2, the compliance in its target coordinates *O-xy* can be calculated as
(6)$$C_{12}^{A}=\sum_{i=1}^{2}T_{i}^{A}C_{i}{\left(T_{i}^{A}\right)}^{T}$$
where $C_{i}^{A}$ is the compliance of flexure hinge *i* with respect to point *A* in target system *A-xy*. *C_i_* is the compliance of flexure hinge *i* in its local coordinates.

Owing to the symmetric structure, the compliance of flexure hinges 3 and 4 in their target coordinates *A-xy* is derived by rotating $C_{12}^{A}$ at 180° around the *y*-axis

(7)$$C_{34}^{A}=R_{y}^{A}(\pi )C_{12}^{A}{\left[R_{y}^{A}(\pi )\right]}^{T}.$$

The parallel connection for a combination of flexure hinges 1 and 2 and assemble hinges 3 and 4 are serially connected to flexure 5. Consequently, the compliance of the lever’s bottom flexure-based part in coordinate *O-xy* can be derived as

(8)$$C_{B}^{O}=T_{A}^{O}{\left[{\left(C_{12}^{A}\right)}^{-1}+{\left(C_{34}^{A}\right)}^{-1}\right]}^{-1}{\left(T_{A}^{O}\right)}^{T}+T_{5}^{O}C_{5}{\left(T_{5}^{O}\right)}^{T}.$$

Hence, the compliance of limb *C* in coordinates *O-xy* is calculated as follows

(9)$$C_{C}^{O}={\left\{{\left(C_{B}^{O}\right)}^{-1}+{\left[T_{6}^{O}C_{6}{\left(T_{6}^{O}\right)}^{T}\right]}^{-1}\right\}}^{-1}+T_{7}^{O}C_{7}{\left(T_{7}^{O}\right)}^{T}.$$

The compliance of limb *C*_1_ in coordinates *O-xy* is obtained by rotating limb *C*’s compliance at 120° around the *z*-axis

(10)$$C_{C_{1}}^{O}=R_{Z}^{O}(\frac{2\pi }{3})C_{H_{1}}^{O}{\left[R_{Z}^{O}(\frac{4\pi }{3})\right]}^{T}.$$

Similarly, the compliance of limb *C*_2_ in coordinates *O-xy* is calculated as follows

(11)$$C_{C_{2}}^{O}=R_{Z}^{O}(\frac{4\pi }{3})C_{H_{1}}^{O}{\left[R_{Z}^{O}(\frac{4\pi }{3})\right]}^{T}.$$

Finally, the compliance of the VPD in output coordinates *O-xy* is obtained as

(12)$$K_{O}={\left(C_{C}^{O}\right)}^{-1}+{\left(C_{C_{1}}^{O}\right)}^{-1}+{\left(C_{C_{2}}^{O}\right)}^{-1}.$$

### 3.2. Input Compliance Modeling

An input compliance model of the VPD is established to calculate the stiffness in the *y*-direction, as shown in Figure 5. Consider that limbs *C*_1_ and *C*_2_ connect in parallel at point *O*. As a result, the compliance of these two limbs is obtained as

(13)$$C_{CC_{1}C_{2}}^{A}=T_{O}^{A}{\left[{\sum_{i=1}^{2}\left(C_{C_{1}}^{A}\right)}^{-1}\right]}^{-1}{\left(T_{O}^{A}\right)}^{-1}.$$

They are connected to the combination of flexure hinges 5, 6, and 7 of limb *C*, which can be calculated as follows

(14)$$\begin{array}{c} C_{CC_{1}C_{2}-765}^{A}={\left\{{\left[C_{CC_{1}C_{2}}^{A}+T_{7}^{A}C_{7}{\left(T_{7}^{A}\right)}^{T}\right]}^{-1}+{\left[T_{6}^{A}C_{6}{\left(T_{6}^{A}\right)}^{T}\right]}^{-1}\right\}}^{-1}+T_{5}^{A}C_{5}{\left(T_{5}^{A}\right)}^{T} \end{array}.$$

Consequently, the VPD’s input stiffness is derived as

(15)$$K_{in}{=C}_{in}^{-1}={\left\{{\left[\sum_{i=1}^{2}T_{i}^{A}C_{i}{\left(T_{i}^{A}\right)}^{T}\right]}^{-1}+{\left[\sum_{j=3}^{4}T_{j}^{A}C_{j}{\left(T_{j}^{A}\right)}^{T}\right]}^{-1}\right\}}^{-1}+{\left(C_{CC_{1}C_{2}-765}^{A}\right)}^{-1}.$$

### 3.3. Amplification Ratio Determination

Larger operating stroke of the VPD is required for various circumstances. Therefore, amplification mechanisms such as the lever model are the most suitable choice. With its merits of fast response and large deformation range, the amplification structure is suitable for magnifying the small input displacements of piezoelectric ceramics. The motion accuracy of the VPD will be significantly disturbed if there is an integration of the lever arm deflection in addition to the hinge stretch effect. A model of the lever used in this analysis is shown in Figure 6. Assuming that elastic deformations are only generated at hinge 6, the link and flexure hinge 7 are rigid without elastic deformations when an input force *F* (the force of PZT) is exerted on point *O*_5_. The relationship of the generalized forces that are generated by hinge 7 (output end of the lever) and hinge 6 can be expressed by
(16)$$\left[\begin{array}{c} M_{6z}\\ F_{6y}\\ F_{6x} \end{array}\right]=\left[\begin{array}{ccc} 1 & l_{3} & -l_{1}\\ 0 & 1 & 0\\ 0 & 0 & 1 \end{array}\right]\left[\begin{array}{c} M_{7z}\\ F_{7y}\\ F_{7x} \end{array}\right]+\left[\begin{array}{c} -l_{2}\\ 0\\ 1 \end{array}\right]F$$
where *l*_1_ and *l*_3_ are the distances of the local coordinates *O*_6_*-xy* and *O*_7_*-xy* in the *x*- and *y*-directions, respectively. *l*_2_ is the input length of the lever.

By defining the force on point *O*_6_ of hinge $F_{6}={\left[\begin{array}{ccc} M_{6z} & F_{6y} & F_{6x} \end{array}\right]}^{T}$ and the related displacement in local coordinate *O*_6_*-xy* as $D_{6}={\left[\begin{array}{ccc} \theta_{6z} & u_{6y} & u_{6x} \end{array}\right]}^{T}$, the following equation is derived based on Hook’s law

(17)$$\left[\begin{array}{c} \theta_{6z}\\ u_{6y}\\ u_{6x} \end{array}\right]=\left[\begin{array}{ccc} C_{\theta_{z}-M_{z}} & C_{\theta_{z}-F_{y}} & 0\\ C_{y-M_{z}} & C_{y-M_{z}} & 0\\ 0 & 0 & 1 \end{array}\right]\left[\begin{array}{c} M_{6z}\\ F_{6y}\\ F_{6x} \end{array}\right].$$

From Equations (16) and (17):(18)$$\left[\begin{array}{c} \theta_{6z}\\ u_{6y}\\ u_{6x} \end{array}\right]=\left[\begin{array}{ccc} C_{\theta_{z}-M_{z}} & C_{\theta_{z}-M_{z}}l_{3}{+C}_{\theta_{z}-F_{y}} & -C_{\theta_{z}-M_{z}}l_{1}\\ C_{y-M_{z}} & C_{y-M_{z}}l_{3}+C_{y-F_{y}} & -C_{y-M_{z}}l_{1}\\ 0 & 0 & C_{x-F_{x}} \end{array}\right]\;\left[\begin{array}{c} M_{7z}\\ F_{7y}\\ F_{7x} \end{array}\right]+\left[\begin{array}{c} -C_{\theta_{z}-M_{z}}l_{2}F\\ -C_{y-M_{z}}l_{2}F\\ C_{x-F_{x}}F \end{array}\right]$$

Since flexure hinge 6 serves as a pivot, the amplification effect at the end of the lever should be considered. The output motion of flexure hinge 7 caused by flexure hinge 6 can be obtained as

(19)$$\left[\begin{array}{c} M_{7z}\\ F_{7y}\\ F_{7x} \end{array}\right]=\left[\begin{array}{ccc} 1 & 0 & 0\\ l_{3} & 1 & 0\\ l_{1} & 0 & 1 \end{array}\right]\left[\begin{array}{c} \theta_{6z}\\ u_{6y}\\ u_{6x} \end{array}\right].$$

Substituting Equation (18) into (19), we obtain the following compliance matrix of the lever output end

(20)$$C_{7}=\left[\begin{array}{ccc} \frac{\partial \theta_{7z}}{\partial M_{7z}} & \frac{\partial \theta_{7z}}{\partial F_{7y}} & \frac{\partial \theta_{7z}}{\partial F_{7x}}\\ \frac{\partial u_{7y}}{\partial M_{7z}} & \frac{\partial u_{7y}}{\partial F_{7y}} & \frac{\partial u_{7y}}{\partial F_{7x}}\\ \frac{\partial u_{7x}}{\partial M_{7z}} & \frac{\partial u_{7x}}{\partial F_{7y}} & \frac{\partial u_{7x}}{\partial F_{7x}} \end{array}\right].$$

The input motion is generated by the PZT force in its local coordinate *A-xy* in the *y*-direction, which can be generated by
(21)$$y_{in}=F\cdot C_{in(22)}$$
where parameter $C_{in}\left(2,2\right)$ is a compliance factor of matrix *C_in_*. From the above, the VPD’s amplification ratio can be obtained as

(22)$$Ramp=\partial u_{7x}/\left[\partial F_{7x}\cdot C_{in(22)}\right]$$

The amplification ratio partially determines the stroke of the VPD.

## 4. Performance Validation and Discussion of the VPD

### 4.1. Testing the Experiment Setup of the VPD

The VPD was manufactured by electrical discharge machining using Aluminum 7075, which has the advantages of higher strength and corrosion resistance. The complete testing experiment setup of the VPD is shown in Figure 7. By amplifying the signal produced from the Power PMAC controller (Delta Tau, Inc.), the power amplifier (E-500, PI, Inc.) was used to actuate the PZTs (model 40vs12 with static stiffness of 35 N/μm, from Harbin Core Tomorrow Science Co., Ltd.) with the consideration of constraint *k_pz_*_t_ > *k_in_*. Then, the transverse motions of the two points (A, B) were obtained.

As shown in Figure 8, as the displacement of the VPD is on the micrometer scale, the output rotation angle *θ_out_* can be calculated as
(23)$$\theta_{\mathrm{out}}\approx \tan\left(\theta_{\mathrm{out}}\right)=\frac{y_{AB}}{l_{AB}}=\frac{y_{B}-y_{A}}{l_{AB}}$$
where *y_A_* and *y_B_* represent the linear strokes of points A and B along the *y*-direction, respectively. The length between the selected measured points A and B is *l_AB_*.

### 4.2. Dynamic Characteristics Evaluation

A type of three-dimensional twenty-node solid element (i.e., C3D20) was selected to mesh the model (Figure 9). In order to enhance the computational accuracy, the HyperMesh software was chosen. The nodes were distributed nonuniformly and concentrated near the hinges. The six fixing holes were constrained in all directions, and the input forces were exerted to each actuating point. In order to evaluate the dynamic performance of the VPD, FEA with the finite element software ABAQUS and the swept excitation method were carried out. The materials and dimensional parameters of the VPD are listed in Table 1. In these parameters, *l*_1_, *l*_2_, and *l*_3_ are related to the maximum amplification ratio [14,21], and *l*_AB_ is a linear negative correlation to the rotational angle [24,25,26,27]. According to the analytical models in [28,29,30], it can be inferred that the frequency of VPD increases nonlinearly with increasing *t* and *w* and decreasing *r*.

The first three modes of the VPD without PZTs were obtained as depicted in Figure 10. The natural frequencies for the first three order mode shapes were 831.12, 852.16, and 1347.8 Hz. The third mode was an in-plane rotational mode that was notably higher than the operating frequency, demonstrating that the proposed VPD could guarantee a reliable rotational motion.

The swept excitation method based on the open-loop experiment was selected to investigate the dynamic performance of the VPD because it is a convenient method. The measured results at point B are displayed in Figure 11. The first three natural frequencies were 817.6, 993.2, and 1270.9 Hz. Accordingly, the first and third measured frequencies at point B coincide with those of the FEA results. However, it should be noted that the second frequency obtained from the swept excitation method is much higher than the result of FEA. The discrepancy is mainly caused by added mass (the sensor brackets as shown in Figure 7). It is clearly seen that both added mass is concentrated on one side, which is ignored in the FEA. This may affect the frequency mainly along one side, while the other sides should not be obvious. Manufacturing errors and imperfect contacts between PZTs and the input ends could also affect the frequencies obtained by tests [31].

### 4.3. Motion Stroke Analysis and Resolution Tests

The closed-loop experiments are conducted in the motion stroke and resolution tests to obtain feedback data. CMD means “command displacement,” while ACT refers to the actual displacement [32]. When investigating the workspace of the developed VPD, it is important to discuss the maximum stroke of the lever output end and the maximum rotational angle. The consecutive step signal was exerted to drive the PZTs (Figure 12a). It can be noticed that the output stroke of the lever at point B could reach a maximum value of 41 μm. A large output stroke is necessary for the flexibility of micropolishing on different scales. According to Equation (23), the maximum output angle of the VPD is graphically displayed in Figure 12b by measuring the displacements at points A and B. It is noted that the maximum output angle was 1.07 mrad. The rotational angle is relatively smaller because (1) the configuration dimension of the center triangular stage is too large; (2) the center shift of the proposed VPD is not well constrained [25].

To realize high-precision motion control, the resolution is the major standard of the VPD. Accordingly, the piezoelectric actuated structure can achieve high resolution. Nevertheless, the flexure-based structure also decreased in resolution due to the influence, for example, the quantization error of the D/A converter and environmental disturbances. A stair-step signal was selected for the PZTs to drive the VPD [26,27]. As shown in Figure 13, the resolution of point B (output end of the lever) and the rotational resolution of the VPD were measured by high-precision capacitive sensors, and were approximately 70 nm and 0.8 μrad, respectively. If the measuring errors and manufacturing tolerances can be further decreased, the VPD has a chance of obtaining comparatively higher resolutions.

### 4.4. Static Performance Analysis

The flexure hinges’ maximum stress should be lower than the selected material’s yield stress. This criteria can keep the VPD from failure during the manufacturing process. As shown in Figure 14, the maximum stress analysis demonstrates that the developed VPD has the capability to realize the maximum output displacement at point B with 46 μm when the PZT forces increase to ${\left[\begin{array}{ccc} 300N & 300N & 300N \end{array}\right]}^{T}$. The corresponding maximum von Mises stresses are 61.12, 33.49, and 26.35 MPa without material failure in squares 1–3, respectively.

The static property of the VPD was verified, i.e., the amplification ratio, input, and rotational stiffness [33]. The corresponding deformation result at the output end of the lever when a force of 180 N was exerted to the input ends is displayed in Figure 15a. And the input–output displacement relationship of the level model was obtained as shown in Figure 15b. Considering the output stroke and the input displacement of the level model, we could obtain a set of amplification ratios directly according to Equation (24). It can clearly be observed that the average of the four sets of amplification ratios is approximately 1.58, which is smaller than the theoretical value of 1.82. The offsets are put down to the fact that the deflections of the lever arms hampered the stroke of the VPD.

The amplification ratio of the level model of the finite element analysis is
(24)$$Ramp^{\prime }=\frac{D_{out}}{D_{in}}$$
where *D*_*out*_ and *D*_*in*_ are the output and input strokes of the level model, respectively.

The generalized definition of stiffness is the ratio of the applied load to the resulting displacement at the action point (compliance is the reciprocal of stiffness). As for the input stiffness, we exerted four sets of input load at the actuators end, and got four sets of corresponding displacement, as shown in Figure 16a. Considering the input load and displacement, we could obtain the input stiffness value which is the reciprocal of the slope of a straight line. Similarly, the output stiffness can be obtained as Figure 16b. The input and output rotational stiffness were evaluated through FEA and were 11.36 N/μm and 2.39 Nm/mrad, respectively. Table 2 lists the results of the analytical modeling and FEA results. The main deviation of the stiffness from the theoretical value with respect to the FEA results come from the center shift of the center triangular stage, as well as the deformation of the links.

## 5. Performance Validation and Discussion of 3D RVMS

### 5.1. Processing Experiment Set Up of 3D RVMS

Considering the practical polishing experiment, the viscous characteristics of diamond abrasives make it easy to agglomerate into large particles and produce severe scratches affecting surface quality. So, we need to reprepare the polishing solution before each experiment. We added a certain amount of dispersant and tested it with a pH tester. Then we used HCl and NaOH solutions to adjust the pH value. In order to control the concentration stability of the slurry, we used an agitator (as shown in Figure 17a) to stir the slurry for 1 h. Then, we homogenized with a high speed homogenizer (as shown in Figure 17b) for 2 h to get the actual polishing diamond slurry. Finally, in the actual experiment, we adopted the method of manually adding the slurry to ensure the processing quality of the SiC workpiece.

To verify the property of the developed 3D RVMS, experiments were conducted using an independent five-axis CNC polishing machine, as shown in Figure 18. The square-shaped SiC ceramics with a size of 10 mm × 10 mm × 2 mm were chosen. The VPD was mounted on a swing station, and the workpiece was fixed on the VPD using bonded wax. The polishing tool was mounted on an X-Z table controlled by air guides. The control signals generated by the Power PMAC were amplified using a power amplifier and were then sent to drive the PEAs. The micropolishing condition is summarized in Table 3.

### 5.2. Experimental Results and Discussion

The SiC ceramic workpieces were ground prior to polishing, as shown in Figure 19. To verify the polishing performance, four reference points were selected from four SiC workpieces, which were based on the average surface roughness (Sa). The surface topographies were measured by an optical surface profiler (ZygoNewview, Middlefield, CT, USA). An enlargement was made of the selected reference points on the original workpiece at a magnification ratio of 50×. The polishing tool was scanned along the *x*-direction, and its rotational speed was controlled to maintain the conditions listed in Table 3. The corresponding processing parameters of the 3D RVMS are summarized in Table 4.

The surface roughness of the polished surface can serve as a sign of the material removal mode. The processing surface morphology with conventional and vibration-assisted polishing methods are shown in Figure 20. It can be seen that both methods have many continuous and discontinuous scratches as well as numerous visible and brittle fracture pits owing to the unstable control of the polishing force and abrasive grains. Considering the practical processing contours, the number of peaks on the surface after conventional polishing, shown in Figure 20a, is higher than that from vibration-assisted polishing, as shown in Figure 20b–d. This indicates that the frequency and rotational angle are sensitive to the surface roughness. This proves the assumption in Section 2 that the surface peaks are improved by introducing rotary vibration assistance. Thus, it can be determined that the removal mode of SiC is changed by adding vibration. Owing to the performance of the 3D RVMS, the surface roughness caused by changing the frequency and rotational angle becomes flexible.

Accordingly, the effect of the rotational angle and operating frequency of the VPD on the polished surface roughness profile is discussed. The detailed data for Sa versus the corresponding parameters are listed in Table 4 and are depicted in Figure 20. Figure 20 shows the maximum Sa in group 1 and the minimum Sa in group 2 of the polished surface, which are observed to be 0.076 μm and 0.037 μm with the corresponding conventional and vibration-assisted polishing, respectively. This demonstrates that the Sa is improved by changing the material removal mode through introducing vibration assistance. Under the rotational angles of group 3 and group 4, the surface roughness of Sa is 0.068 and 0.037 μm at rotational angles of 0.17 mrad and 0.39 mrad, respectively, at the same frequency. These results indicate that a large rotational angle can effectively enhance the surface quality.

However, the distinction of the surface roughness which was obtained by changing the operating frequency was relatively smaller, as shown in group 2 and group 4 (Figure 21). Owing to the restrictions of the experimental environmental conditions, there was no three-channel signal generator, which made an open-loop processing experiment impossible to conduct.

## 6. Conclusions

A novel micropolishing system combining the simple characteristics of the CNC precision processing machine and the multiple-DOF property of the piezoelectrically actuated VPD was proposed. A static analysis was performed by compliance modeling via the matrix method. After prototype fabrication, the dynamic and static performances of the VPD were discussed using FEA. Through open-loop and closed-loop testing experiments and closed-loop processing experiments, the 3D RVMS was determined to satisfy the requirements for precision polishing. The major conclusions can be summarized as follows:(1)The amplification ratio and the input/output compliances of the VPD were analytically modeled based on the matrix method. The theoretical results had good agreement with the analytical and FEA results.(2)Through experimental tests, the maximum rotational angle of the VPD could reach 1.07 mrad with an output resolution of 0.8 µrad, while the maximum stroke at point B was 41 µm with a resolution of 70 nm. The VPD was also capable of achieving high first natural frequencies, which were examined to be 817.6 Hz at point B.(3)Compared with a nonvibration CNC polishing system, Sa was clearly improved by the proposed 3D RVMS. According to the contrast polishing experiments of the VPD’s rotational angle, Sa is vulnerable to a large rotational angle. The results of the processing experiments validated the authenticity and flexibility of the proposed vibration-assisted micropolishing principle.

## Figures and Tables

**Figure 1 micromachines-10-00502-f001:**
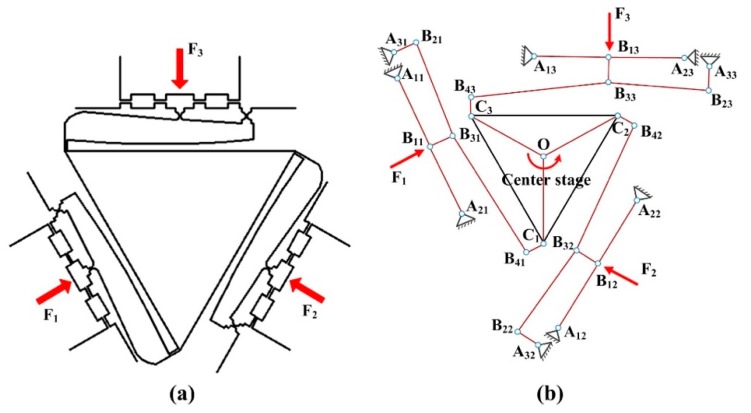
(**a**) Major part and (**b**) pseudo-rigid-body model of the vibration-assisted polishing device (VPD).

**Figure 2 micromachines-10-00502-f002:**
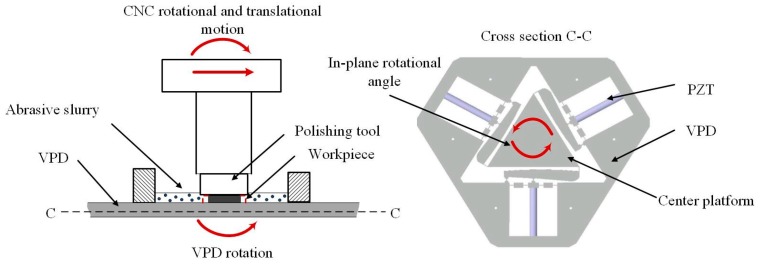
Configuration of the 3D rotary vibration-assisted micropolishing system (RVMS).

**Figure 3 micromachines-10-00502-f003:**
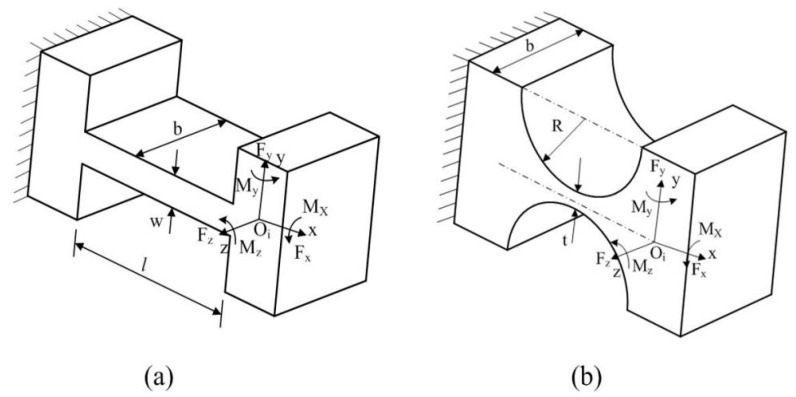
Coordinate system of the (**a**) right-angle hinge and (**b**) right-circular hinge.

**Figure 4 micromachines-10-00502-f004:**
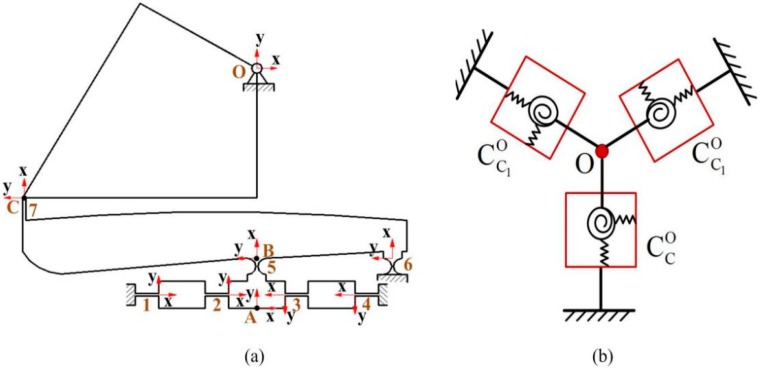
(**a**) Third of kinematic chains and (**b**) parallel spring model of the VPD.

**Figure 5 micromachines-10-00502-f005:**
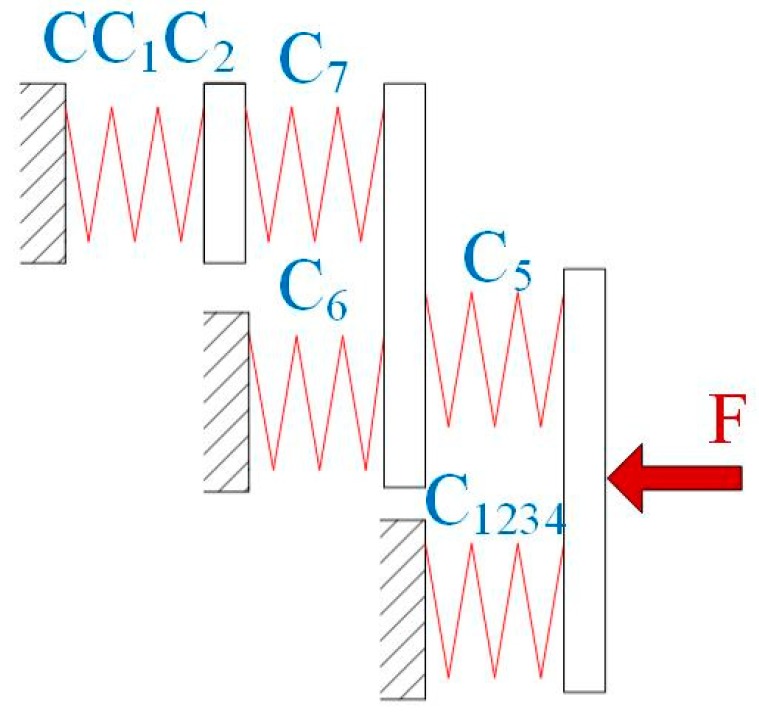
Stiffness model of the VPD with one limb actuated.

**Figure 6 micromachines-10-00502-f006:**
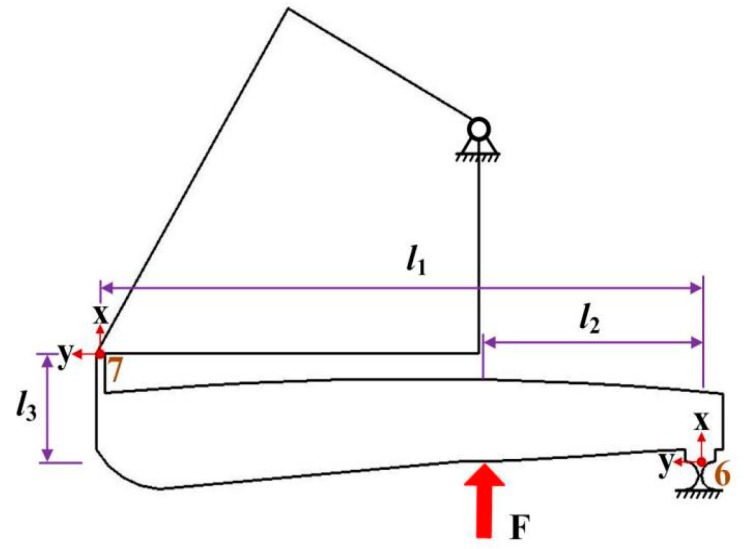
Schematic diagram of the lever mechanism.

**Figure 7 micromachines-10-00502-f007:**
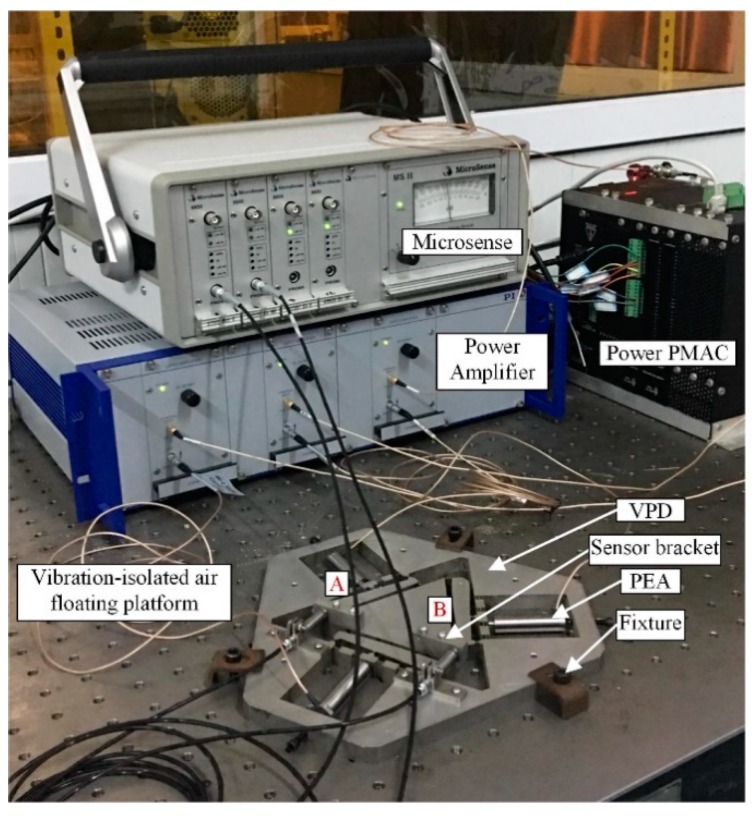
Testing experiment setup of the VPD.

**Figure 8 micromachines-10-00502-f008:**
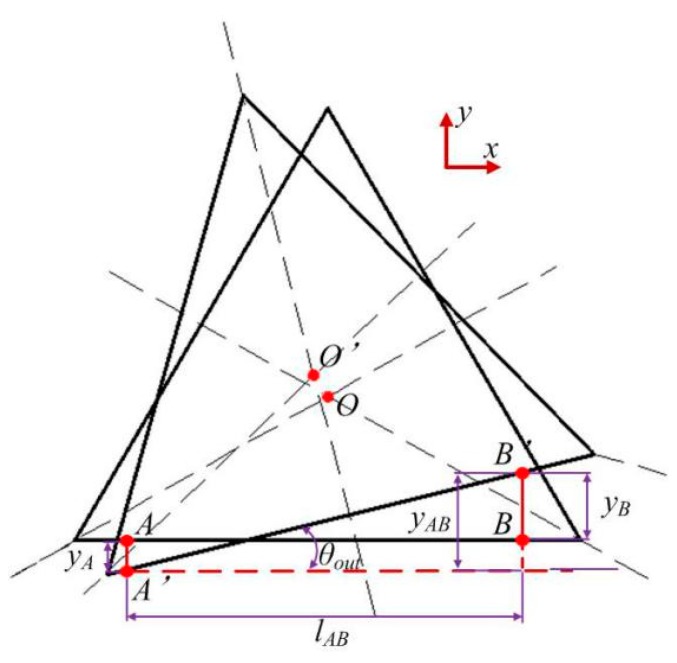
Schematic diagram of the computation of rotational angle.

**Figure 9 micromachines-10-00502-f009:**
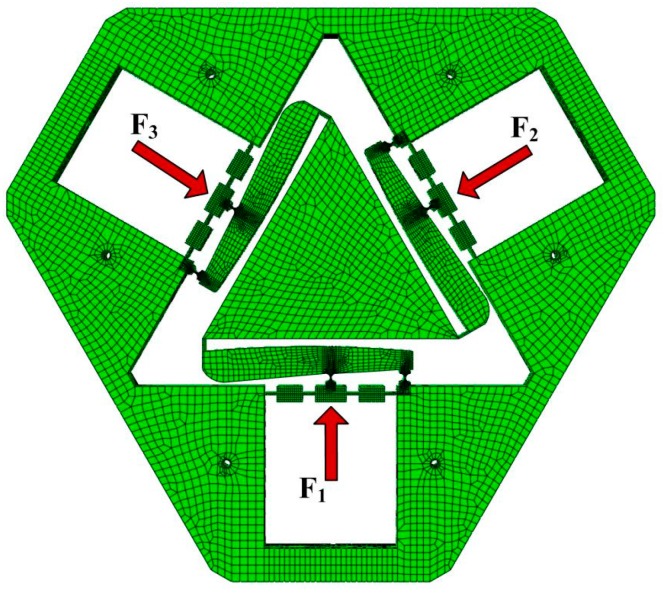
Finite-element model of the vibration-assisted polishing device (VPD).

**Figure 10 micromachines-10-00502-f010:**
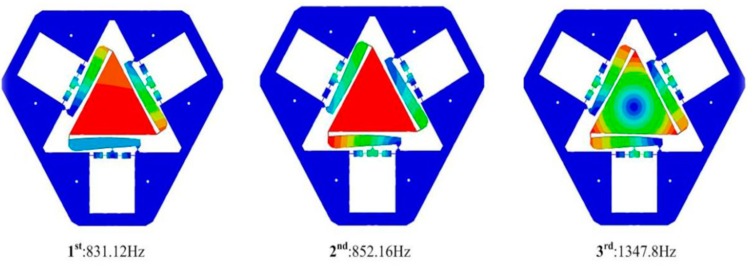
Mode shapes of the developed VPD.

**Figure 11 micromachines-10-00502-f011:**
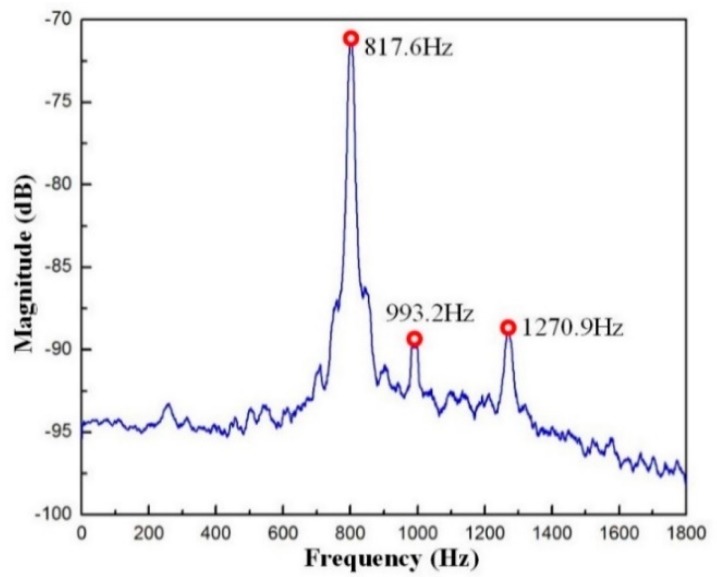
Dynamic responses of the VPD at point B.

**Figure 12 micromachines-10-00502-f012:**
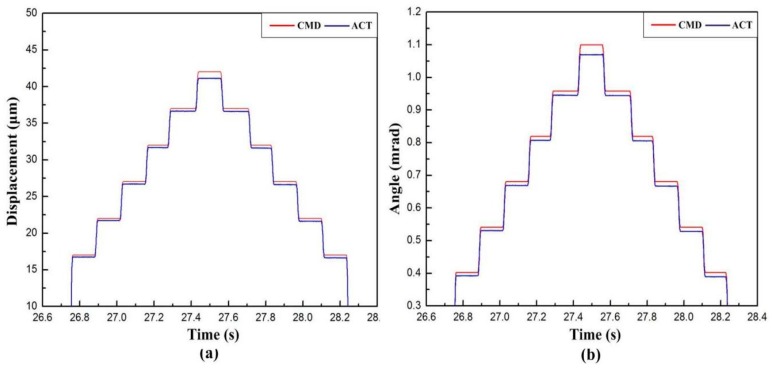
Experimental results of the (**a**) motion stroke of the lever output end and (**b**) rotational angle of the VPD.

**Figure 13 micromachines-10-00502-f013:**
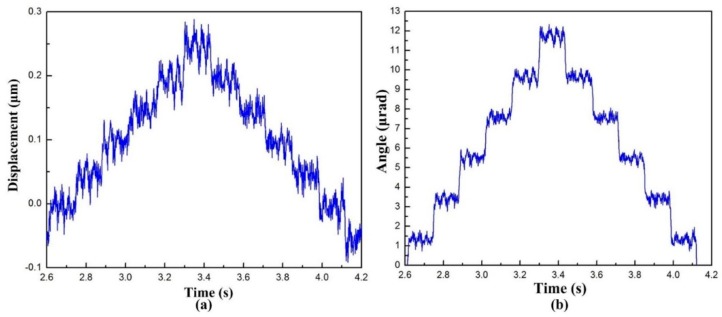
Experimental resolutions of (**a**) the output end of the lever and (**b**) the output center triangular stage of the VPD.

**Figure 14 micromachines-10-00502-f014:**
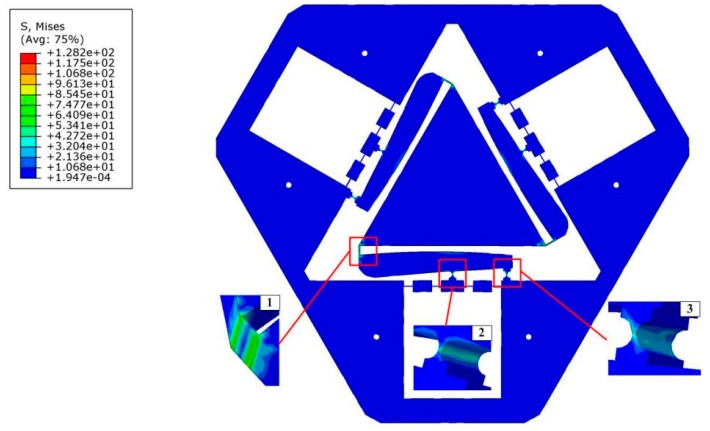
Stress simulation of the VPD.

**Figure 15 micromachines-10-00502-f015:**
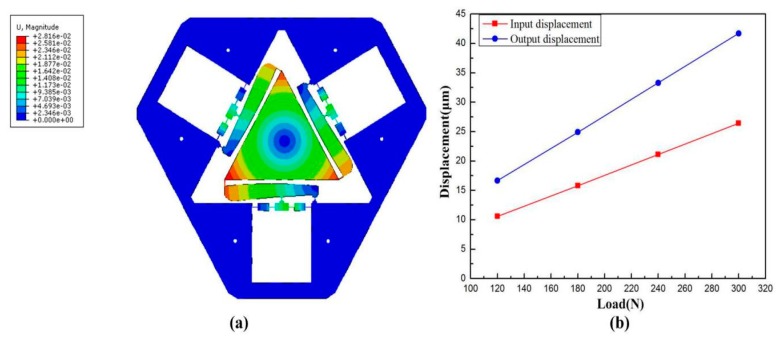
(**a**) Deformation behavior and (**b**) the input–output displacement relationship of the level model.

**Figure 16 micromachines-10-00502-f016:**
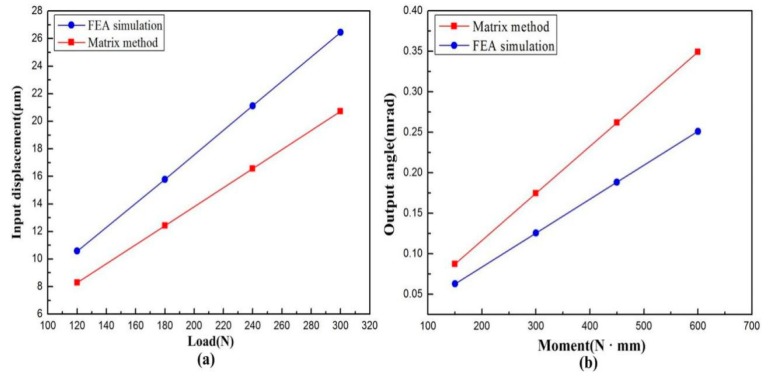
Force-displacement curves of (**a**) input displacement and (**b**) output angle.

**Figure 17 micromachines-10-00502-f017:**
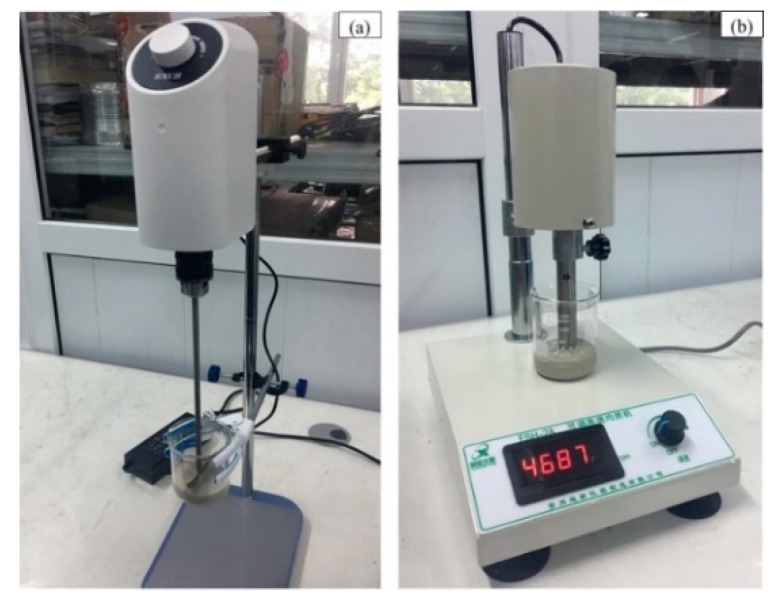
Stability control device of the diamond slurry. (**a**) Agitator and (**b**) high speed homogenizer.

**Figure 18 micromachines-10-00502-f018:**
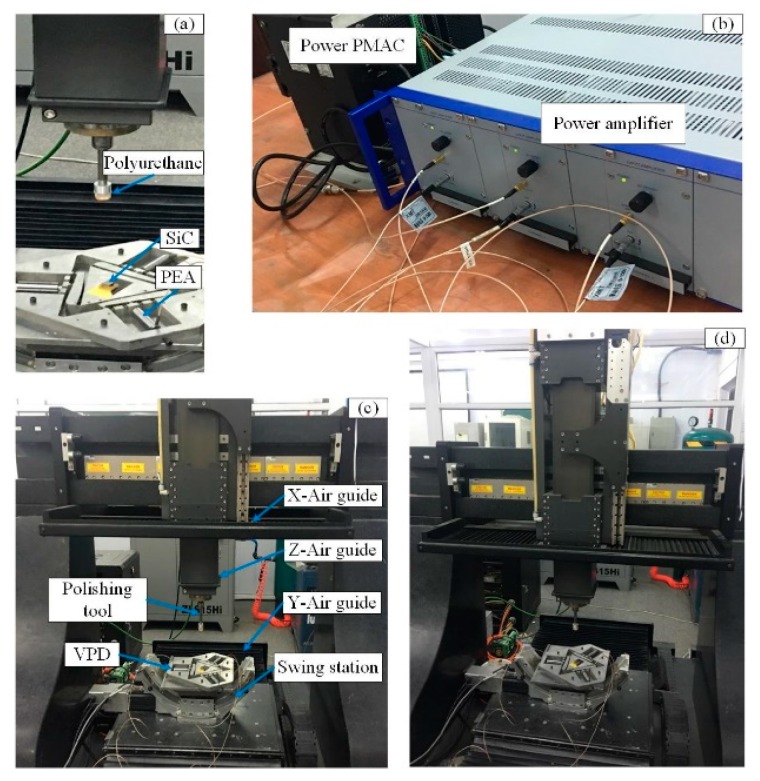
(**a**) Processing part, (**b**) signal control part, (**c**) the major part of the processing setup and (**d**) processing experiment setup of 3D RVMS.

**Figure 19 micromachines-10-00502-f019:**
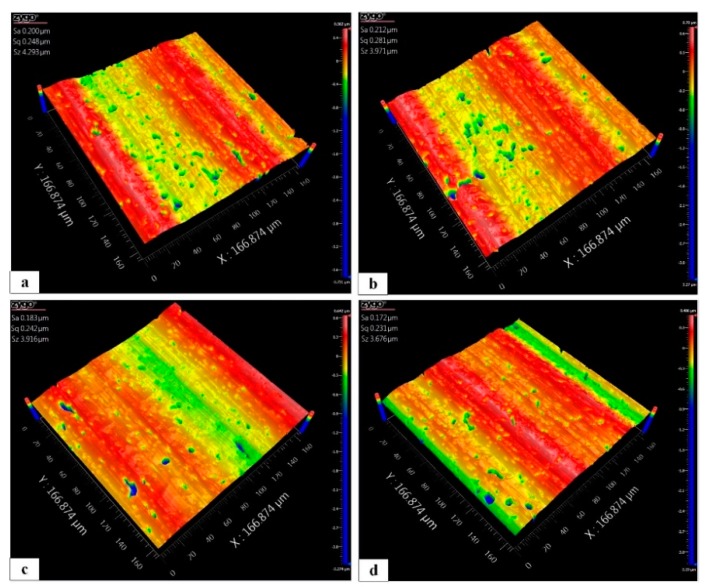
Workpiece surface morphology of four reference points: (**a**) workpiece 1, (**b**) workpiece 2, (**c**) workpiece 3 and (**d**) workpiece 4.

**Figure 20 micromachines-10-00502-f020:**
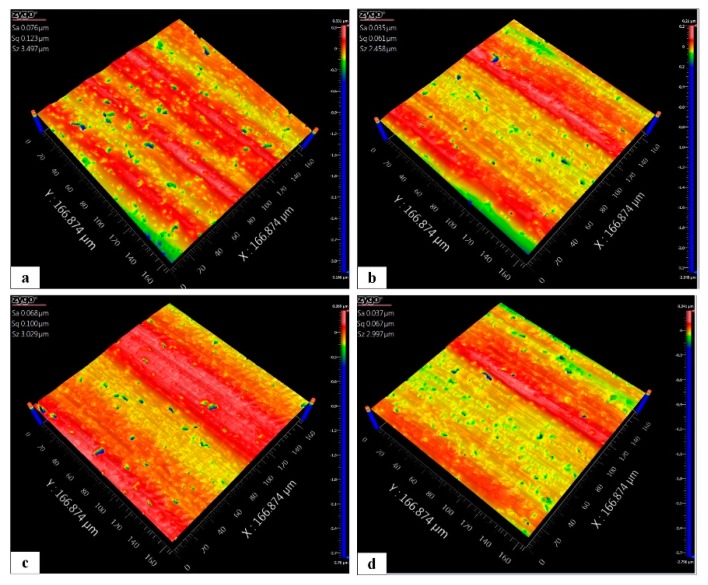
Workpiece surface morphology: (**a**) nonvibration polishing (workpiece 1), (**b**) vibration polishing (workpiece 2) (**c**) small-rotational-angle polishing (workpiece 3) and (**d**) large-rotational-angle polishing (workpiece 4).

**Figure 21 micromachines-10-00502-f021:**
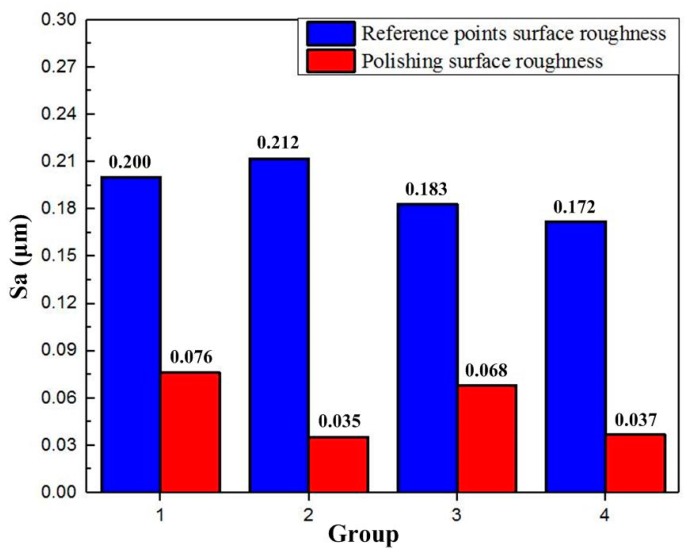
Data of average surface roughness (Sa) for polishing.

**Table 1 micromachines-10-00502-t001:** Main parameters of the VPD.

***r* (mm)**	***b* (mm)**	***t* (mm)**	***w* (mm)**	***l_AB_* (mm)**	***l_1_* (mm)**
0.82	12	0.4	0.6	88	81
***l_2_* (mm)**	***l_3_* (mm)**	***E* (GPa)**	***σ* (MPa)**	***μ***	***ρ* (kg/m^3^)**
25	21	71.7	503	0.33	2810

**Table 2 micromachines-10-00502-t002:** Performances evaluated by the analysis model and finite element analysis (FEA) results.

Performance	Input Stiffness (N/μm)	Output Compliance (Nm/mrad)	Amplification Ratio
**Matrix model**	14.49	1.72	1.82
**FEA**	11.36	2.39	1.58
**Deviation (%)**	21.6	28.0	13.2

**Table 3 micromachines-10-00502-t003:** Micropolishing Experiment condition.

Workpiece	SiC Ceramic
**Number**	4
**Rotational speed**	1200 r/min
**Translational speed**	3 mm/s
**Polishing tool**	Polyurethane
**Radius curvature**	8 mm
**Abrasive**	Diamond slurry
**Grain size**	3.0 μm
**Polishing time**	1 h

**Table 4 micromachines-10-00502-t004:** The corresponding processing parameters of the 3D RVMS.

Workpiece Number	Piezo-Electric (PZT) Actuators Frequency (Hz)	Output Displacement (μm)	Rotational Angle (mrad)
**1**	×	×	×
**2**	80	23	0.39
**3**	50	11	0.17
**4**	50	23	0.39

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
