# Peer review of "Development of a Novel Three Degrees-of-Freedom Rotary Vibration-Assisted Micropolishing System Based on Piezoelectric Actuation"

_micromachines, 2019, doi:10.3390/mi10080502_

Round 1

Reviewer 1 Report

The authors proposed a novel 3 DOF rotary vibration-assisted micropolishing system. But it is not clear enough that why we need 3 DOF. The limit of the current technology is not clearly described. Is it really helpful to use 3 DOF?

Besides,  improvement about the polishing process with 3 DOF vibration as compared to the traditional vibration polishing cannot bee seen from the polishign results. Pls explain.

What is the concentration of the slurry. And how to control the concentration stability of the slurry which is very important to obtain good surface quality, since no slurry circulation system was shown in the figure17.

Author Response

1. The authors proposed a novel 3 DOF rotary vibration-assisted micropolishing system. But it is not clear enough that why we need 3 DOF.

Author response:

Thanks for the reviewer’s suggestion. We should make an apology for the failure to clearly interpret why we need 3 degrees of freedom (DOF). We explain them as follows:

The 3 DOF in the article means the proposed vibration-assisted polishing device is driven by three piezo-electric (PZT) actuators. And the piezoelectrically actuated vibration-assisted micro-polishing method could enable the workpiece to move along the rotational direction because of this driving mode. Compare with the current vibration-assisted processing technology which is focus on using 2 DOF actuators to drive the non-resonant vibration-assisted device that generate trajectory with orthogonal actuators or parallel actuators, the proposed actuated mode can provide a new rotating motion mode (related to 3 DOF actuated) that could change the material removal mode and enhance the surface quality.

Author action:

In order to clear demonstrate why we need 3 DOF, we have modified abstract according to the response to reviewer.

2. The limit of the current technology is not clearly described. Is it really helpful to use 3 DOF?

Author response:

Thanks for the reviewer’s suggestion. We sincerely apologize for the contribution of using 3 DOF that did not correctly convey to the reviewer. We reinterpreting the innovation of our work and the limit of the current technology as follows:

Our purpose is to design a novel 3-DOF piezoelectrically actuated vibration-assisted polishing device enable the workpiece to move along the rotational direction. Most scholars who study vibration-assisted processing technology are focus on generating trajectory with orthogonal actuators or parallel actuators. Specific reference of these current technology is described as follows:

(1) Zhiwei Zhu, Member, IEEE; Suet To, Member, IEEE

He proposed 2-DOF fast-tool-servo-assisted diamond turning to enable the cutting tool to move along two directions with decoupled motions, which used two parallel actuators.

The reference: Zhu. Z, Zhou. X, Liu. Z, Wang. R, and Zhu. L, “Development of a piezoelectrically actuated two- degree-of-freedom fast tool servo with decoupled motions for micro-/nanomachining,” Precis. Eng, vol. 38, no. 4, pp. 809-820, 2014, 10.1016/j.precisioneng.2014.04.009.

(2) Wu-Le Zhu, Member, IEEE; Kornel F. Ehmann, Fellow, ASME

He presented a novel 2-DOF vibration-assisted compliant cutting system to be used on conventional machines for generating textured surfaces, which used two parallel actuators.

The reference: Zhu. W, Zhu. Z, He. Y, Ehmann. K. F, Ju. B, and Li. S, “Development of a novel 2-D vibration-assisted compliant cutting system for surface texturing,” IEEE-ASME. T. Mech, vol. 22, no. 4, pp. 1796-1806, 2017, 10.1109/TMECH.2017.2693996.

(3) Wanqun Chen

He investigated the tool-workpiece separation mechanism of 2-DOF assisted milling process by introducing two orthogonal actuators.

The reference: Chen. W, Zheng. L, Xie. W, Yang. K, and Huo, D, “Modelling and experimental investigation on textured surface generation in vibration-assisted micro-milling,” J. Mater. Process. Tech, vol. 266, pp. 339-350, 2019, 10.1016/ j.jmatprotec.2018.11.011.

Our team also designed vibration-assisted roll-type polishing (EARP) system considering the two orthogonal actuators to extend the capabilities of existing mechanisms for precision polishing.

The reference: Gu. Y, Chen. X, Lin. J, Lu. M, and Lu. F, “Vibration-Assisted Roll-Type Polishing System Based on Compliant Micro-Motion Stage,” Micromachines, vol. 9, no. 10, pp. 499, 2018, 10.3390/mi91 00499.

    2. Compare to the travel scale of most resonant vibration-assisted devices are micrometer, our   device can deliver maximum rotational angle in mrad scale because of 3-DOF piezoelectrically actuating. It is true that most resonant mechanisms are suffering the drawbacks of small vibration amplitudes. Specific reference is as follows:

(1) H. Suzuki, Member, CIRP

He proposed an ultrasonic vibration assisted polishing machine and the small polishing tool is vibrated at an ultrasonic frequency with amplitude 10μm.

The reference: Suzuki. H, Moriwaki. T, Okino. T, and Ando. Y, “Development of ultrasonic vibration assisted polishing machine for micro aspheric die and mold,” CIRP. Ann, vol. 55, no. 1, pp. 385-388, 2006, 10.1016/S0007-8506(07)60441-7.

He proposed an ultrasonic two-axis vibration assisted polishing machine with axial vibration amplitude 40 μm and flexural vibration amplitude 30μm.

The reference: Suzuki. H, Hamada. S, Okino. T, Kondo. M, Yamagata. Y, and Higuchi. T,

“Ultraprecision finishing of micro-aspheric surface by ultrasonic ultrasonic two-axis vibration assisted polishing,” CIRP. Ann, vol. 59, no. 1, pp. 347-350, 2010, 10.1016/j.cirp.2010.03.117.

(2) Qingliang Zhao

He investigated the material removal mechanism of ultrasonic vibration assisted polishing (UVAP) on micro cylindrical SiC surface, which used vibration amplitude 1.0-4.0 μm in the experiment.

The reference: Zhao. Q, Sun. Z, and Guo. B, “Material removal mechanism in ultrasonic vibration assisted polishing of micro cylindrical surface on SiC,” Int. J. Mach. Tool. Manu, vol. 103, pp. 28-39, 2016, 10.1016/ j.ijmachtools.2016.01.003.

(3) Zhiqiang Liang

He investigated the material removal characteristics in elliptical ultrasonic assisted grinding (EUAG) of monocrystal sapphire that used the vibrator was constructed by bonding a piezoelectric ceramic device (PZT) with very low vibration amplitude 1μm.

 The reference: Liang. Z, Wu. Y, Wang. X, and Zhao. W, “A new two-dimensional ultrasonic assisted grinding (2D-UAG) method and its fundamental performance in monocrystal silicon machining,” Int. J. Mach. Tool. Manu, vol. 50, no. 8, pp. 728-736, 2010, 10.1016/j.ijmachtools.2010. 04.005.

Author action:

In order to clear describe the limit of the current technology and why really 3 DOF is helpful, we have modified the introduction according to the response to reviewer.

3. Besides, improvement about the polishing process with 3 DOF vibration as compared to the traditional vibration polishing cannot bee seen from the polishign results. Pls explain.

Author response:

Thanks for the reviewer’s suggestion. We sincerely apologize for the failure to clearly interpret the contrast between the polishing process with 3 DOF vibration and the traditional vibration polishing. We reinterpreting the contrast as follows:

In order to verify the authenticity and validity of the comparison, I hope the reviewer will understand that I consider the conventional vibration-assisted polishing as non-resonant vibration-assisted polishing actuated by 2-DOF parallel actuators to verify the contrast with the results of the polishing process with 3-DOF vibration. Because such method are conducted on the same independent five-axis CNC machine and used same material SiC ceramic workpiece. The contrast details as follow:

(Article detail: Gu, Y.; Zhou, Y.; Lin, J.; Yi, A.; Kang, M.; Lu, H.; Xu, Z. Analytical Prediction of Subsurface Damages and Surface Quality in Vibration-Assisted Polishing Process of Silicon Carbide Ceramics. Materials 2019, 12, 1690. Materials 2019, 12(10), 1690; https://doi.org/10.3390/ma12101690 (first author))

During this conventional vibration-assisted polishing experiment, the surface roughness of conventional polishing (without vibration-assisted) was 220nm and the proposed polishing method driven by 2-DOF parallel actuators was 73nm. Due to the limitations of the output stroke and motion mode, it is clearly seen that the 3D surface topography is not uniformity and the productivity is low. Compare with the non-resonant vibration-assisted polishing actuated by 2-DOF parallel actuators, the proposed polishing process with 3-DOF vibration can achieve surface roughness 37nm with high productivity because the large output stroke. Besides, the surface topography is uniformity because the surface peaks are improved by introducing rotary vibration assistance.

4. What is the concentration of the slurry. And how to control the concentration stability of the slurry which is very important to obtain good surface quality, since no slurry circulation system was shown in the figure17.

Author response:

Thanks for the reviewer’s comment. The original diamond slurry is obtained from Chinese company, according to the instructions we know it contain 10-15% diamond abrasive grains. Considering the practical polishing experiment, the viscous characteristics of diamond abrasives make it easy to agglomerate into large particles and produce severe scratches affecting surface quality. So we need to reprepare the polishing solution before each experiment. We added a certain amount of dispersant and tested it with a pH tester. Then we used HCl and NaOH solutions to adjust pH value. In order to control the concentration stability of the slurry, we used an agitator (as shown in figure 17(a)) to stir the slurry with 1 hour. Then, we homogenized with a high speed homogenizer (as shown in figure17 (b)) for 2 hours to get the actual polishing diamond slurry. Finally, in the actual experiment, we adopt the method of manually adding the slurry to ensure the processing quality of the SiC workpiece. Owing to the restrictions of the experimental environmental conditions, there was no slurry circulation system. We sincerely apologize for the inconvenience caused to the reviewer and reader. Thanks to the reviewer’s suggestion. We will add the slurry circulation system to practical experiments in the future.

Author action:

In order to clear explain the suggestion from reviewer, we have modified the 5.1. Processing experiment set up of 3D RVMS according to the response to reviewer.

5. Moderate English changed required.

Author response:

Thanks for the reviewer’s suggestion. We embellished the article according to the reviewer’s request.

Reviewer 2 Report

1. Please recheck the journal abbreviation in Refs. 12 and 19 to fit the request of the journal.

2.Please recheck the grammar for the sentences in page 6 (lines 172-173), page 15 (lines 346-347) and page 15 (lines 356-357).

Author Response

1. Please recheck the journal abbreviation in Refs. 12 and 19 to fit the request of the journal.

Author response:

Thanks for the reviewer’s suggestion. We revised the article according to the reviewer’s request.

2. Please recheck the grammar for the sentences in page 6 (lines 172-173), page 15 (lines 346-347) and page 15 (lines 356-357).

Author response:

Thanks for the reviewer’s suggestion. We revised the article according to the reviewer’s request.

3. Moderate English changed required.

Author response:

Thanks for the reviewer’s suggestion. We embellished the article according to the reviewer’s request.

Round 2

Reviewer 1 Report

I'm fine with the revision.

Add subtitles of Fig 17.

Reviewer 2 Report

NO